# Betel Nut Chewing Is Associated with the Risk of Kidney Stone Disease

**DOI:** 10.3390/jpm12020126

**Published:** 2022-01-18

**Authors:** Chun-Kai Chang, Jia-In Lee, Chu-Fen Chang, Yung-Chin Lee, Jhen-Hao Jhan, Hsun-Shuan Wang, Jung-Tsung Shen, Yao-Hsuan Tsao, Shu-Pin Huang, Jiun-Hung Geng

**Affiliations:** 1Department of Urology, Kaohsiung Municipal Siaogang Hospital, Kaohsiung 812, Taiwan; kai05010501@gmail.com (C.-K.C.); leeyc12345@yahoo.com.tw (Y.-C.L.); ghostdeityj@gmail.com (J.-H.J.); whs524@gmail.com (H.-S.W.); uro.shenjt@gmail.com (J.-T.S.); paranoid289@gmail.com (Y.-H.T.); 2Department of Urology, Kaohsiung Medical University Hospital, Kaohsiung Medical University, Kaohsiung 80756, Taiwan; shpihu73@gmail.com; 3Department of Psychiatry, Kaohsiung Medical University Hospital, Kaohsiung Medical University, Kaohsiung 80756, Taiwan; u9400039@gmail.com; 4Department of Physical Therapy, Tzu Chi University, Hualien 97004, Taiwan; cfchang711@mail.tcu.edu.tw; 5Department of Urology, Faculty of Medicine, College of Medicine, Kaohsiung Medical University, Kaohsiung 80756, Taiwan; 6Graduate Institute of Clinical Medicine, College of Medicine, Kaohsiung Medical University, Kaohsiung 80756, Taiwan; 7Ph.D. Program in Environmental and Occupational Medicine, College of Medicine, Kaohsiung Medical University, Kaohsiung 80756, Taiwan; 8Research Center for Environmental Medicine, Kaohsiung Medical University, Kaohsiung 80756, Taiwan

**Keywords:** epidemiologic study, kidney stone disease, nephrolithiasis, betel nut, areca nut, risk factors

## Abstract

(1) Background: Betel nut chewing injures bodily health. Although, the relationship between betel nut chewing and kidney stone disease (KSD) is unknown. (2) Methods: We analyzed 43,636 men from Taiwan Biobank. We divided them into two groups on the status of betel nut chewing, the never-chewer and ever-chewer groups. Self-reported diagnosed KSD was defined as the subject’s medical history of KSD in the questionnaire. Logistic regression was used to analyze the association of betel nut chewing and the risk of KSD. (3) Results: The mean age of subjects in the present study was 50 years, and 16% were ever-chewers. KSD was observed in 3759 (10.3%) and 894 (12.6%) participants in the group of never-chewer and ever-chewer groups, respectively. Higher risk of KSD was found in participants with betel nut chewing compared with to without betel nut chewing (odds ratio (OR), 1.094; 95% confidence interval (95% CI), 1.001 to 1.196). Furthermore, the daily amounts of betel nut chewing >30 quids was associated with a more than 1.5-fold increase (OR, 1.571; 95% CI, 1.186 to 2.079) in the odds of KSD; (4) Conclusions: Our study suggests that betel nut chewing is associated with the risk of KSD and warrants further attention to this problem.

## 1. Introduction

Kidney stone disease (KSD), also known as nephrolithiasis, is a common urological problem, resulting in a clinical burden and high cost to health care systems. Annual medical expenditures for KSD in the United States was above $2.1 billion in 2000 [1]. If the stone blocks the urinary tract, it may cause several symptoms such as severe pain in the flanks, gross hematuria, and vomiting, with the obstruction further resulting in hydronephrosis, postrenal azotemia, acute kidney injury and sepsis. Furthermore, KSD is a risk factor for chronic kidney disease, cardiovascular disease and osteoporosis [2,3,4,5]. Preventing the occurrence of KSD is an increasingly important issue and stone formation has been proven as related to genetic and environmental factors including age, gender, body mass index (BMI), diet, fluid intake, caffeine, smoking, type 2 diabetes mellitus (DM) and climate [6,7,8,9,10,11]. However, there are other risk factors that have not yet been studied, including betel nut.

Betel nut, also called areca nut, is associated with several diseases such as oral ulcers, periodontal disease and cancers of the oral cavity and esophagus [12,13,14]. Furthermore, it affects many systems of the human body, including the cardiovascular system, digestive system, immune system, endocrine system, nervous system and renal system [12]. Some studies have revealed that chewing betel nut could injure the kidney [15,16,17], but few studies have demonstrated a relationship between chewing betel nut and KSD. The aim of the present study is to explore the relationship between betel nut chewing and KSD.

## 2. Materials and Methods

### 2.1. Taiwan Biobank and Study Design

The Taiwan Biobank (TWB), a population-based biobank, consists of a cohort of more than 100,000 participants with no cancer diagnosis at baseline. The majority of participants are classed racially as Han Chinese (over 99%). TWB collects the information of lifestyles, health risk factors, medical history, physical examination, blood tests and other biospecimens, with this data providing causes and mechanisms of common diseases to researchers. By combining medical information and disease-causing factors, TWB aims to facilitate better treatment and prevention and improve the health of people [18,19,20,21].

On this basis, the data from TWB (2008 to 2019) was used to examine the association between betel nut chewing and KSD in men. A total of 43,848 men were enrolled in the present study as shown in Figure 1. Participants with missing information concerning nut experience (N = 62), age (N = 1), smoking status (N = 9), alcohol status (N = 41), past history of dyslipidemia (N = 11), body mass index (N = 49), physical activity status (N = 16), status of marriage (N = 13), and status of education (N = 10) were excluded, so 43,636 men were entered into the final analysis. All participants signed the informed consent, and all investigations followed the Declaration of Helsinki. The present study was approved by the Institutional Review Board of Kaohsiung Medical University Hospital (KMUHIRB-E(I)-20190398).

### 2.2. Betel Nut Chewing Assessments

All participants were asked the following question: “Have you ever had experience of betel nut chewing.” Participants saying “Never or only had once or twice” were assigned as the never-chewer group, those replying “Yes” were assigned as the ever-chewer group. Among the participants with experience of betel nut chewing, the following questions were then asked: “When did you start chewing betel nuts?” “Are you a current chewer?” “How many betel nuts do you have each day?” According to the frequency of betel nut use, we divided participants into “never-chewer”, “≤10 betel nuts per day”, “11–30 betel nuts per day”, and “>30 betel nuts per day.”

### 2.3. Self-Reported Diagnosed KSD

KSD was defined by a self-reporting history of diagnosed KSD (yes/no), and the detailed questions were as below: “Have you ever had history of diagnosed KSD?” If participants replied “yes”, they were further asked “When were you diagnosed with kidney stones?” The interviewers would repeat these questions to ensure that all participants understood and answered consistently.

### 2.4. Statistical Methods

Volunteers in this study were divided into never-chewer and ever-chewer groups. Categorial variables are presented as percentages and continuous variables as mean ± standard deviation. Pearson χ2 test was performed to examine the differences among categorical variables, independent *t*-tests examined the differences among continuous variables, while logistic regression was used to analyze the association of betel nut chewing and self-reported diagnosed KSD. To examine the dosage effect between frequency of betel nut use and KSD, a subgroup of 39,113 participants with adequate information was analyzed, being divided into 4 groups as described previously, with logistic regression being conducted to identify the association between frequency of betel nut use and KSD in this subgroup. We used R version 3.6.2 and SPSS 20.0 to perform the analyses, with a *p* value < 0.05 regarded as statistically significant for all analyses.

## 3. Results

### 3.1. Clinical Characteristics of the Study Participants

Of the 43,636 men, the mean age was 50 ± 11 years, with 36,521 men (84%) in the never-chewer group and 7115 men (16%) in the ever-chewer group (Table 1). Men with betel nut chewing tended to be older, with higher blood pressure, a higher alcohol consumption rate, a higher smoking rate, a lower educational level, higher white blood cell counts, higher platelet counts, higher serum hemoglobin, total cholesterol, triglycerides, blood sugar, uric acid, creatinine, and a higher BMI than those in the never-chewer group (Table 1).

### 3.2. Betel Nut Chewing Was Associated with an Increasing Risk of KSD

A total of 7728 men had self-reported diagnosed KSD in the present study, with 6824 (6%) in the never-chewer group and 904 (12%) in the ever-chewer group. In univariable binary logistic analysis, subjects with higher BMI, smoking experience, alcohol consumption, physical activity, history of hypertension, DM, dyslipidemia, gout, and betel nut experience had higher odds of KSD. On the contrary, men with higher albumin had lower odds of KSD. Participants in the ever-chewer group were associated with a 1.25-fold increase of KSD than those in the never-chewer group (odds ratio (OR), 1.252; 95% confidence interval (95% CI), 1.159 to 1.354) (Table 2). In a subgroup analysis for subjects without a history of hypertension, DM, dyslipidemia, gout, and obesity (BMI ≥ 30 kg/m^2^), the subjects in the ever-chewer group were still associated with a higher prevalence of KSD (OR, 1.140; 95% CI, 1.014 to 1.283) (Appendix A).

After adjusting for cohort effect (using a threshold of a *p* value < 0.05 from the univariate analysis (Table 2)), including age, BMI, smoking status, alcohol status, physical activity, marital status, education status, systolic blood pressure, diastolic blood pressure, past history of hypertension, past history of DM, past history of dyslipidemia, past history of gout, platelet counts, serum albumin, fasting glucose, hemoglobin A1c, total cholesterol, triglyceride, high-density lipoproteins (HDL) cholesterol, low-density lipoproteins (LDL) cholesterol, creatinine, and uric acid, subjects in the ever-chewer group were significantly associated with a higher risk of KSD than those in the never-chewer group (OR, 1.094; 95% CI, 1.001 to 1.196) (Table 3).

### 3.3. Dose-Response Effect between Betel Nut Chewing and the Risk of KSD

To further examine the association between the amounts of daily betel nut use and KSD, a subgroup of participants with adequate information was collected. In multivariate logistic regression analysis, subjects with ≤10 betel nuts per day, 11–30 betel nuts per day, and >30 betel nuts per day had odds ratios of 1.056 (95% CI, 0.863 to 1.291), 1.045 (95% CI, 0.870 to 1.255) and 1.571 (95% CI, 1.186 to 2.079) in the risk of KSD compared with those in the never-chewer group (Table 4). The odds increased slightly at the lower amounts of exposure (≤10 betel nuts per day and 11–30 betel nuts per day), then increased significantly at the higher amounts of exposure, which demonstrated the dose-response relationship between the amounts of daily betel nut use and the risk of KSD.

## 4. Discussion

In this cross-sectional study of a large-scale, community-based representative population in Taiwan, betel nut chewing was significantly associated with KSD after adjustment for confounders. We also found that the more betel nuts chewed daily, the higher the prevalence of KSD. To the best of our knowledge, this is the first large study to demonstrate this association to date.

Betel nut, the seed of the Areca catechu palm tree, is common in Southern, South East Asian and Pacific islands countries [12]. The prevalence of chewing betel nut is estimated as above 10% of the world population [22]. Betel nut is one of the most common addictive natural products on earth [23]. The pernicious effect of betel nut is not just restricted to the oral cavity but also affects systemic health including the central nervous system, cardiovascular disease, metabolic syndrome, type 2 DM, hypertension, dyslipidemia, and chronic kidney disease [16,24,25,26,27,28,29]. A previous study enrolled eight patients with recurrent urinary stones and discovered the possibility of increased risk of urinary stones in betel nut chewers [30]. There was no control group in this study, which limited the ability to evaluate the odds of KSD between betel nut chewers and non-chewers. Liu et al. conducted a cross-sectional case-control study and found that current betel nut chewers had a higher risk for calcium urolithiasis than non-chewers (OR, 1.97; 95% CI, 1.06 to 3.64) [31]. However, this was a single hospital study with a small cohort (354 cases vs. 354 age- and sex-matched controls), which limited the generalizability of their findings. Our study builds on the results of both studies and confirms the association between betel nut chewing and the risk of KSD by a large population-based study.

A strength of our research includes the finding of the dose-response effect between betel nut chewing and the risk of KSD. We observed that subjects with high doses of exposure (>30 betel nuts per day) had higher risk of KSD compared with those with low dose exposure. Similar dose-response effects could be found between betel nut consumption and other diseases such as cardiovascular disease [32], metabolic disease [32], and oral cancer [33]. A meta-analysis, which consisted of 17 studies with 388,134 patients, evaluated the impact of chewing betel nut on cardiovascular disease, metabolic disease, and all-cause mortality, with results showing that betel nut not only increased the risk of events but demonstrated significant dose-response relationships as well [32]. Another case-control study examined the dose-response relationships between the risk of oral cancer and lifetime cumulative exposure of betel nut, and observed that the risk of oral cancer rose steeply at low amounts and plateaued at higher exposure to betel nut [33]. In line with these studies, we also observed dose-response effects between betel nut chewing and the risk for KSD.

The mechanism that links chewing betel nut with KSD remains unclear, although the potential mechanisms might be related to the renal damage caused by arecoline, which is a major component of alkaloid in betel nut. Previous studies have shown that arecoline could increase the oxidative stress and cause DNA damages in vitro and in vivo [34,35,36]. Furthermore, Hsieh et al. found that arecoline could promote morphological changes and migration in human kidney (HK2) cells and could induce epithelial-mesenchymal transition (EMT) and subsequent renal fibrosis by upregulating the expression of N-cadherin, vimentin, α-SMA and collagen [37]. These findings suggest that arecoline is associated with renal dysfunction and can cause chronic injury to the kidney [37]. An animal study also showed that rats fed with betel nuts had higher percentages of kidney tubular injury than those without [38]. Kidney stone formation is proposed to be related to the damage of renal tubule epithelial cells and renal fibrosis [39,40,41]; additionally, increased oxidative stress plays a major role in the deposition of urinary crystals in the renal tubules during stone formation [42]. Taken together, betel nut and its components may increase oxidative reaction, cause renal injury and eventually, stone formation.

According to the International Agency for Cancer Research (IARC) and the specialized cancer agency of the World Health Organization (WHO), betel nut has been classified as a group 1 carcinogen. Betel nut is also associated with multiple health-harming effects, including the kidney [29,43]. Unfortunately, despite the adverse effects of betel nut, there is a lack of global policy to control the use of betel nuts [44].

The Taipei city government, the capital of Taiwan, once proposed a regulation: “Betel Nut Hygiene Management Autonomy Provision,” which aimed to prohibit chewing betel nuts in public places, but it failed eventually due to opposition from betel nut users, growers, and retailers [45]. Until now, it can only add warnings on the packaging of betel nuts and educate the public about the hazards of chewing betel nuts, but any supposed beneficial effects still need further research [46]. Through our study, we hope to remind the government and the public about the potential harm of betel nuts and further reduce demand.

To the best of our knowledge, this study is the first, large-scale, multiple-covariate and community-based one with more than 43,000 subjects investigating the association between betel nut chewing and KSD. Additionally, the frequency of betel nut use per day was also analyzed to investigate the dose-response effect. Despite these strengths, there are some limitations. Firstly, the major limitation is the use of self-reported diagnosed KSD instead of medical records or image-confirmed KSD; however, in such large-scale research, it is difficult to review each and every medical record and image. Such alternative methods have been widely used in large studies, such as the UK biobank and Biobank Japan. In addition, the self-reported diagnosed KSD by the subjects represents the clinical significance of the disease, rather than asymptomatic or incidental diseases. Secondly, a causal relationship could not be confirmed in this cross-sectional study. Thirdly, all the subjects come from Taiwan, and no other people in different countries were included, which might limit the generalizability of our findings. Fourthly, future studies are required to include other dietary habits or daily fluid-amount intake for evaluation.

## 5. Conclusions

Our study demonstrates that not only hypertension, DM, dyslipidemia, gout, obesity but also betel nut chewing are strongly associated with KSD. With an increase in the daily intake of betel nut, the risk of KSD also increased. This implies that reducing the population of betel nut users could improve men’s health and further attention to this problem is warranted.

## Figures and Tables

**Figure 1 jpm-12-00126-f001:**
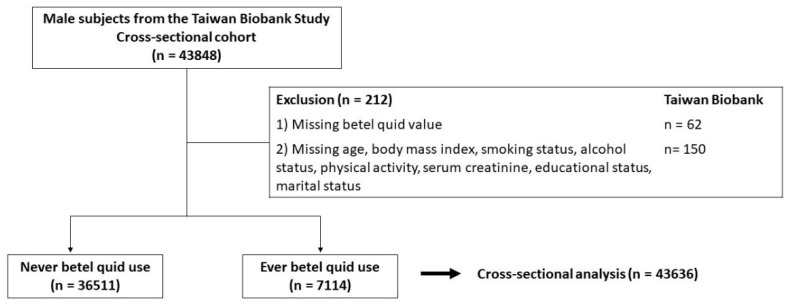
Study participants were classified by status of betel nut.

**Table 1 jpm-12-00126-t001:** Clinical characteristics of the study participants classified by betel nut use.

Characteristics	Total (N = 43,636)	Betel Nut Use		*p* Value
		Never, N = 36,521	Current or Ex-Chewer, N = 7115	
**Demographic Data**				
Age, years	50 ± 11	50 ± 12	51 ± 10	<0.001
BMI, kg/m^2^	25.4 ± 3.6	25.2 ± 3.5	26.1 ± 3.8	<0.001
Alcohol status, ever, n (%)	8156 (19)	5021 (14)	3135 (44)	<0.001
Smoking status, ever, n (%)	25,036 (57)	18,290 (50)	6746 (95)	<0.001
Physical activity, yes, n (%)	18,490 (42)	15,906 (44)	2584 (36)	<0.001
Married, yes, n (%)	37,735 (87)	31,271 (86)	6464 (91)	<0.001
Education status, n (%)				<0.001
≤Elementary	1312 (3)	882 (2)	430 (6)	
Middle-to-high school	13,202 (30)	9213 (25)	3989 (56)	
≥College	29,122 (67)	26,426 (73)	2696 (38)	
SBP, mm Hg	126 ± 17	126 ± 17	127 ± 17	<0.001
DBP, mm Hg	78 ± 11	78 ± 11	80 ± 11	<0.001
**Comorbidities**				
Hypertension, n (%)	7333 (17)	5817 (16)	1516 (21)	<0.001
Dyslipidemia, n (%)	4053 (9)	3168 (8)	885 (12)	<0.001
Diabetes mellitus, n (%)	2960 (7)	2312 (6)	648 (9)	<0.001
Gout, n (%)	4230 (10)	3429 (9)	801 (11)	<0.001
**Laboratory data**				
White blood counts, 10^9^/L	6.0 ± 1.7	6.0 ± 1.6	6.4 ± 1.8	<0.001
Red blood counts, 10^12^/L	5.1 ± 0.5	5.1 ± 0.5	5.1 ± 0.5	0.469
Platelet, 10^9^/L	227 ± 54	226 ± 54	229 ± 56	<0.001
Hb, g/dL	15.1 ± 1.2	15.1 ± 1.2	15.2 ± 1.3	<0.001
Albumin, g/dL	4.6 ± 0.2	4.6 ± 0.2	4.6 ± 0.2	<0.001
Fasting glucose, mg/dL	99 ± 23	99 ± 22	102 ± 29	<0.001
HbA1c, %	5.8 ± 0.9	5.8 ± 0.9	6.0 ± 1.0	<0.001
Total cholesterol, mg/dL	192 ± 35	192 ± 35	193 ± 38	0.070
TG, mg/dL	138 ± 118	132 ± 100	168 ± 182	<0.001
HDL, mg/dL	48 ± 11	48 ± 11	46 ± 11	<0.001
LDL, mg/dL	122 ± 32	122 ± 31	120 ± 33	0.004
Creatinine, mg/dL	0.9 ± 0.4	0.9 ± 0.3	0.9 ± 0.5	0.134
UA, mg/dL	6.4 ± 1.4	6.4 ± 1.3	6.6 ± 1.4	<0.001

Abbreviations: BMI: body mass index; SBP: systolic blood pressure; DBP: diastolic blood pressure; Hb: hemoglobin; HbA1c: hemoglobin A1c; TG: triglyceride; HDL: high-density lipoproteins cholesterol; LDL: low-density lipoproteins cholesterol; UA: uric acid.

**Table 2 jpm-12-00126-t002:** Association between parameters and KSD in univariable binary logistic analysis (N = 43,636).

Parameters	Odds Ratio (95% CI)	*p*
Age (per 1 year)	1.033 (1.030 to 1.036)	<0.001
BMI (per 1 unit)	1.049 (1.040 to 1.057)	<0.001
Smoking status, ever (vs. never)	1.259 (1.183 to 1.340)	<0.001
Alcohol status, ever (vs. never)	1.101 (1.020 to 1.188)	0.013
Physical activity, yes (vs. no)	1.129 (1.062 to 1.201)	<0.001
Married, yes (vs. no)	2.262 (2.013 to 2.543)	<0.001
Education status, ≤elementary school (vs. others)	0.808 (0.766 to 0.852)	<0.001
SBP (per 1 unit)	1.010 (1.009 to 1.012)	<0.001
DBP (per 1 unit)	1.017 (1.014 to 1.019)	<0.001
Hypertension, yes (vs. no)	2.360 (2.203 to 2.527)	<0.001
Diabetes mellitus, yes (vs. no)	1.726 (1.558 to 1.912)	<0.001
Dyslipidemia, yes (vs. no)	1.989 (1.824 to 2.169)	<0.001
Gout, yes (vs. no)	1.832 (1.680 to 1.998)	<0.001
White blood counts (per 10^9^/L)	1.012 (0.994 to 1.030)	0.180
Red blood counts (per 10^12^/L)	0.975 (0.917 to 1.037)	0.421
Platelet (per 10^9^/L)	0.998 (0.998 to 0.999)	<0.001
Hb (per 1 unit)	1.022 (0.996 to 1.048)	0.093
Albumin (per 1 unit)	0.661 (0.582 to 0.752)	<0.001
Fasting glucose (per 1 unit)	1.004 (1.003 to 1.005)	<0.001
HbA1c (per 1 unit)	1.126 (1.094 to 1.159)	<0.001
Total cholesterol (per 1 unit)	0.999 (0.998 to 1.000)	0.002
TG (per 1 unit)	1.000 (1.000 to 1.001)	<0.001
HDL (per 1 unit)	0.989 (0.986 to 0.992)	<0.001
LDL (per 1 unit)	0.999 (0.998 to 0.999)	0.003
Creatinine (per 1 unit)	1.168 (1.103 to 1.237)	<0.001
UA (per 1 unit)	1.051 (1.028 to 1.074)	<0.001
Betel nut status, ever (vs. never)	1.252 (1.159 to 1.354)	<0.001

Abbreviations: as Table 1 and CI: confidence interval.

**Table 3 jpm-12-00126-t003:** Association between parameters and KSD in multivariate binary logistic analysis (N = 43,636).

Parameters	Odds Ratio (95% CI)	*p*
Age (per 1 year)	1.026 (1.022 to 1.030)	<0.001
Body mass index (per 1 unit)	1.031 (1.021 to 1.041)	<0.001
Smoking status, ever (vs. never)	1.117 (1.043 to 1.196)	0.002
Alcohol status, ever (vs. never)	0.910 (0.837 to 0.989)	0.026
Physical activity, yes (vs. no)	0.927 (0.868 to 0.991)	0.026
Married, yes (vs. no)	1.471 (1.297 to 1.669)	<0.001
Education status, ≤elementary school (vs. others)	0.990 (0.934 to 1.050)	0.748
SBP (per 1 unit)	0.993 (0.990 to 0.996)	<0.001
DBP (per 1 unit)	1.015 (1.011 to 1.019)	<0.001
Hypertension, yes (vs. no)	1.588 (1.466 to 1.720)	<0.001
Diabetes mellitus, yes (vs. no)	1.128 (0.990 to 1.285)	0.070
Dyslipidemia, yes (vs. no)	1.288 (1.171 to 1.418)	<0.001
Gout, yes (vs. no)	1.390 (1.265 to 1.527)	<0.001
White blood counts (per 10^9^/L)	-	-
Red blood counts (per 10^12^/L)	-	-
Platelet (per 10^9^/L)	0.999 (0.999 to 1.000)	0.039
Hb (per 1 unit)	-	-
Albumin (per 1 unit)	1.065 (0.923 to 1.228)	0.388
Fasting glucose (per 1 unit)	1.000 (0.998 to 1.002)	0.818
HbA1c (per 1 unit)	0.967 (0.912 to 1.025)	0.261
Total cholesterol (per 1 unit)	0.996 (0.992 to 0.998)	0.013
TG (per 1 unit)	1.001 (1.000 to 1.001)	0.053
HDL (per 1 unit)	1.001 (0.996 to 1.006)	0.623
LDL (per 1 unit)	1.004 (1.000 to 1.007)	0.037
Creatinine (per 1 unit)	1.064 (0.985 to 1.034)	0.065
UA (per 1 unit)	1.009 (0.985 to 1.034)	0.468
Betel nut status, ever (vs. never)	1.094 (1.001 to 1.196)	0.047

Abbreviations: as Table 1 and CI: confidence interval. Adjusting for age, smoking status, alcohol status, physical activity, marital status, educational status, body mass index, SBP, DBP, history of hypertension, history of dyslipidemia, history of diabetes mellitus, history of gout, platelet count, HbA1c, serum fasting glucose, total cholesterol, TG, LDL, HDL, serum albumin, creatinine, and UA.

**Table 4 jpm-12-00126-t004:** Relative risk for KSD according to frequency of betel nut use (N = 39,113).

Betel Nut Frequency	No. of Cases (%)	Number at Risk	Adjusted Odds Ratio (95% CI)	*p* Value
Never chewer	3759 (10)	36,521	1.000 (Reference)	-
≤10 betel nut per day	119 (12)	974	1.056 (0.863 to 1.291)	0.598
11–30 betel nut per day	151 (12)	1242	1.045 (0.870 to 1.255)	0.639
>30 betel nut per day	66 (18)	376	1.571 (1.186 to 2.079)	0.002

Abbreviations: as Table 1 and CI: confidence interval. Adjusting for age, smoking status, alcohol status, physical activity, marital status, educational status, body mass index, SBP, DBP, history of hypertension, history of dyslipidemia, history of diabetes mellitus, history of gout, platelet count, HbA1c, serum fasting glucose, total cholesterol, TG, LDL, HDL, serum albumin, creatinine, and UA.

## Data Availability

Restrictions apply to the availability of these data. Data was obtained from Taiwan Biobank and are available with the permission of Taiwan Biobank.

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
