# Peer review of "Betel Nut Chewing Is Associated with the Risk of Kidney Stone Disease"

_jpm, 2022, doi:10.3390/jpm12020126_

Round 1

Reviewer 1 Report

Authors of the present study aimed at investigating the relationship between  betel nut chewing and kidney stone disease (KSD). I congratulate with the authors. The issue is original, the methodology is correct and the manuscript is well written. I simply suggest to improve in the "Introduction" section the quality of the following sentence by better clarifying the aim of the study: " The aim of the present study is therefore to elaborate this issue."

Reviewer 2 Report

The authors describe the first large population study assessing the association between betel nut chewing and risk of kidney stone disease, based on self reporting of a history of kidney stone disease. 

The article is clearly written and methods and results are well presented. The references are comprehensive. 

I have two concerns:

1) Betel nut chewing is a predictor of kidney stone disease on multivariate analysis but the lower limit of the 95% confidence interval is only just above 1 and the hazard ratio is lower than that of hypertension, BMI, Diabetes, Gout and dyslipidemia. I think you should state this in your conclusion, i.e., betel nut chewing is a predictor of KSD, but it appears less strong than these other variables.

2) The above listed variables are all potential confounding variables for betel nut chewing. This should be mentioned in the conclusion. Please can you attempt to address this by looking at a subgroup of the study cohort that did not have any of the above diseases and demonstrating an association between betel nut chewing and KSD in this subgroup.
